# BUB1 Inhibition Overcomes Radio- and Chemoradiation Resistance in Lung Cancer

**DOI:** 10.3390/cancers16193291

**Published:** 2024-09-27

**Authors:** Shivani Thoidingjam, Sushmitha Sriramulu, Oudai Hassan, Stephen L. Brown, Farzan Siddiqui, Benjamin Movsas, Shirish Gadgeel, Shyam Nyati

**Affiliations:** 1Department of Radiation Oncology, Henry Ford Health, Detroit, MI 48202, USA; 2Department of Surgical Pathology, Henry Ford Cancer Institute, Henry Ford Health, Detroit, MI 48202, USA; 3Henry Ford Health + Michigan State University Health Sciences, Detroit, MI 48202, USA; 4Department of Radiology, Michigan State University, East Lansing, MI 48824, USA; 5Division of Hematology/Oncology, Department of Medicine, Henry Ford Health, Detroit, MI 48202, USA

**Keywords:** BUB1, lung cancer, chemotherapy, radiotherapy, chemoradiation, molecular target

## Abstract

**Simple Summary:**

Elevated levels of the mitotic checkpoint kinase BUB1 are frequently observed in most solid cancers, including lung cancer. High BUB1 expression is linked to poorer patient survival. In the present study, we aimed to evaluate if a BUB1 targeting agent could enhance the effectiveness of chemotherapy and chemoradiation in lung cancer. Our findings suggest that BUB1 inhibitor sensitizes lung cancer cell lines to chemotherapy, radiotherapy, and chemoradiation and thus validate BUB1 as a novel molecular target.

**Abstract:**

**Background**: Despite advances in targeted therapies and immunotherapies, traditional treatments like microtubule stabilizers (paclitaxel, docetaxel), DNA-intercalating platinum drugs (cisplatin), and radiation therapy remain essential for managing locally advanced and metastatic lung cancer. Identifying novel molecular targets could enhance the efficacy of these treatments. **Hypothesis**: We hypothesize that BUB1 (Ser/Thr kinase) is overexpressed in lung cancers and its inhibition will sensitize lung cancers to chemoradiation. **Methods**: BUB1 inhibitor (BAY1816032) was combined with cisplatin, paclitaxel, a PARP inhibitor olaparib, and radiation in cell proliferation and radiation-sensitization assays. Biochemical and molecular assays evaluated the impact on DNA damage signaling and cell death. **Results**: Immunostaining of lung tumor microarrays (TMAs) confirmed higher BUB1 expression in non-small cell lung cancer (NSCLC) and small cell lung cancer (SCLC) compared to normal tissues. In NSCLC, BUB1 overexpression correlated directly with the expression of TP53 mutations and poorer overall survival in NSCLC and SCLC patients. BAY1816032 synergistically sensitized lung cancer cell lines to paclitaxel and olaparib and enhanced cell killing by radiation in both NSCLC and SCLC. Molecular analysis indicated a shift towards pro-apoptotic and anti-proliferative states, evidenced by altered BAX, BCL2, PCNA, and Caspases-9 and -3 expressions. **Conclusions**: Elevated BUB1 expression is associated with poorer survival in lung cancer. Inhibiting BUB1 sensitizes NSCLC and SCLC to chemotherapies (cisplatin, paclitaxel), targeted therapy (olaparib), and radiation. Furthermore, we present the novel finding that BUB1 inhibition sensitized both NSCLC and SCLC to radiotherapy and chemoradiation. Our results demonstrate BUB1 inhibition as a promising strategy to sensitize lung cancers to radiation and chemoradiation therapies.

## 1. Introduction

Lung cancer is the leading cause of cancer-related deaths worldwide and approximately a quarter million new cases and half that number of deaths are expected in 2024 in the United States alone [1]. Lung cancer is categorized into two primary subtypes, non-small cell (NSCLC) and small cell (SCLC). NSCLC includes histological subtypes including lung adenocarcinoma (LUAD), lung squamous cell carcinoma (LUSC), and large cell carcinoma (LCLC) and constitutes 85% of cases while SCLC accounts for the remaining 15% [2]. Despite advances in targeted therapy and immunotherapy, microtubule stabilizers (paclitaxel, docetaxel), DNA-intercalating platinum agents (cisplatin), and radiation therapy continue to play a critical role in the management of locally advanced and metastatic NSCLC and SCLC. Both subtypes pose significant treatment challenges including late-stage diagnoses and resistance to various therapies [2]. The average 5-year survival rate for lung cancer is 25% [1]. DNA repair inhibitors such as PARP inhibitors (PARPi, olaparib) are being clinically evaluated for their cytotoxic activities. However, in a recent trial, olaparib maintenance failed in platinum-sensitive advanced NSCLC patients (PIPSeN trial). Several clinical trials evaluated targeted agents and checkpoint inhibitors in adjuvant settings for early-stage NSCLC [3,4] and relapsed SCLC [5] with limited success. Therefore, identifying molecular drivers of primary and adaptive chemo-radiotherapy resistance that can be effectively targeted represents a critical unmet clinical need. Developing multimodal treatment approaches [6] that can improve the efficiencies of standard of care (SOC) treatment will be crucial for lung cancer patient management in the future [6,7,8,9]. 

BUB1 is a highly conserved serine/threonine kinase that plays a critical role in cell division [10] and in DNA damage response [11,12]. Dysregulated BUB1 expression is implicated in various malignancies, including lung cancers [13,14,15]. Our previous research has shown that BUB1’s kinase activity drives aggressive cancer phenotypes through TGF-β signaling [16,17]. BUB1 inhibition reduced tumor xenograft growth [14,18] and sensitized cancer cells to taxanes, ATR, or PARP inhibitors, confirming BUB1 as a therapeutic target for enhancing the efficacy of chemotherapies and targeted therapies [19]. Recently, we identified a role for BUB1 in mediating radioresistance in TNBC [20]. However, the role of BUB1 in improving the effectiveness of radiotherapy or chemoradiation in lung cancers, and in particular SCLC, remains unexplored. Here, we demonstrate that BUB1 inhibition sensitizes both NSCLC and SCLC to radiotherapy and chemoradiation, thus providing a rationale for designing clinical trials combining BUB1i with SOC therapies. 

## 2. Materials and Methods

### 2.1. BUB1 Expression in Lung Cancer

BUB1 expression was assessed in the Lung Cancer Explorer (LCE) [21] and UALCAN [22] databases. The TCGA_LUAD_2016, TCGA_LUSC_2016, and Rousseaux_2013 datasets from LCE [21] were used to generate BUB1 expression and survival data. 

### 2.2. Immunohistological Analysis of BUB1 Expression in Lung Tumor Microarrays

Formalin-fixed, paraffin-embedded NSCLC and SCLC TMAs were procured from Tissue Array (https://www.tissuearray.com, Derwood, MD, USA). The TMAs (LUC1201MSur 120 cases and SCLC681Sur 68 cases) were immunostained with an IHC-verified BUB1 antibody (1:50 dilution; Abcam, MA, USA, #ab195268). Staining intensity and distribution were manually scored by a pathologist who was blind to the clinical parameters.

### 2.3. Cell Culture

NSCLC (A549, H2030, H1975, Calu-1) and SCLC (NCI-H2198, and NCI-H1876) cell lines were obtained from ATCC (Manassas, VA, USA). These cells carry different driver mutations (Appendix A). Cell lines were cultured in DMEM (ATCC, 30-2002), McCoy (ATCC, 30-2002), RPMI-1640 (ATCC, 30-2001) or HITES media (ATCC, 30-2006) which were supplemented with 5–10% Fetal Bovine Serum (Thermo Fisher Scientific, FB12999102, Waltham, MA, USA) and Penicillin–Streptomycin (5000 U/mL) (Thermo Fisher Scientific, 15070063). Cells were maintained at 37 °C with 5% CO_2_ and were regularly tested for Mycoplasma contamination (MycoAlert PLUS; Lonza. LT07-705, Lexington, MA, USA).

### 2.4. Drug Treatment and Radiation 

BUB1 inhibitor BAY1816032 (CT-BAY181), paclitaxel (CT-0502), and olaparib (AZD2281, CT-A2281) were obtained from Chemietek (Indianapolis, IN, USA), and cisplatin (PHR1624-200MG) was sourced from Millipore Sigma (Burlington, MA, USA). DNAPK inhibitor NU7441 (S2638) was acquired from Selleck Chemicals (Houston, TX, USA). Stock solutions of BAY1816032 (20 mM), paclitaxel (10 mM), cisplatin (20 mM), and olaparib (20 mM) were made in DMSO, aliquoted, and stored at −80 °C. Working solutions with varying drug concentrations were freshly prepared in culture media to maintain drug stability and activity during assays. A CIX3 cabinet irradiator [Xstrahl Inc., Suwanee, GA, USA] equipped with a 320 kV metal ceramic X-ray tube and operating at 10 mA was used for radiation exposures. A 1 mm copper filter was used for beam hardening. Cell monolayers in plastic 6-well tissue culture plates were exposed to 2, 4, and 6 Gy radiation. Quality assurance testing included a calibrated and certified small field electrometer and routine use of X-ray-sensitive EBT3 GAFCHROMIC film (Ashland Specialty Ingredients, Bridgwater, NJ, USA).

### 2.5. Cell Survival Assay

Cells were seeded at 1000 cells per well in a 96-well plate. After 24 h, cells were treated with BAY1816032, cisplatin, paclitaxel, and olaparib at concentrations ranging from nanomolar to micromolar in triplicates. Cells were treated for 3 days or 7 days. Cell proliferation was assessed using the AlamarBlue^®^ Cell Viability Reagent (Thermo Fisher Scientific, Cat. No. DAL1100) according to the manufacturer’s protocol. All the experiments were repeated at least 3 times. Data were analyzed using GraphPad Prism V8 (Boston, MA, USA), and *p*-values were calculated to determine the significance.

### 2.6. Clonogenic Cell Survival Assay

The cells were seeded in 6-well tissue culture plates at different densities in triplicates. After 24 h, cells were treated with several drug concentrations (BAY1816032, paclitaxel, cisplatin, and olaparib), alone or in combinations and irradiated (0, 2, 4, and 6 Gy) an hour after drug treatment. Cells were allowed to grow for 10–15 days until colonies with 50 or more cells formed. Colonies were fixed, stained, and counted to calculate plating efficiency (PE), survival fractions (SF), and radiation enhancement ratios (rERs). The experiments were performed at least three times.

### 2.7. DNA Damage Assay

DNA damage was assessed by quantitating cells with γH2AX foci as described in [20]. Briefly, cells were seeded on cover glasses and allowed to attach for 24 h before treating with 1 µM BAY1816032. After 1 h, cells were irradiated (4 Gy) and fixed at different time points for assessing DNA double-strand breaks. Cells were stained with anti-phospho-Histone H2A.X (Ser139) antibody (Millipore Sigma, Cat. No. 05-636-25UG) and counterstained with Alexa Fluor 568-tagged secondary antibody (Invitrogen, Cat. No. A-11004, Carlsbad, CA, USA). Nuclei were counterstained with DAPI, and images were captured using a fluorescent microscope (Zeiss, Oberkochen, DE-BW, Germany), with an average of 100 nuclei per image. Cells with more than 10 foci were counted as being positive for γH2AX. At least 3 random fields/samples were counted. Each experiment was repeated at least three times.

### 2.8. Quantitative Real-Time PCR

For quantitative real-time polymerase chain reaction (qPCR) studies, 1 × 10^5^ cells were seeded in 6-well plates. After 24 h, the cells were treated with 1 μM BUB1i or DNAPKi for 72 h. Total RNA was extracted using TRIzol reagent (Thermo Fisher Scientific, 15596026) and RNA concentration and purity were assessed using Nanodrop (Thermo Fisher Scientific, Nanodrop 2000c). First-strand cDNA synthesis was performed using SuperScript III Reverse Transcriptase (Thermo Fisher Scientific, Cat. No. 18080044), dNTPs (Thermo Fisher Scientific, Cat. No. R0191), and random primers (Thermo Fisher Scientific, Cat. No. 48190011). qPCR reactions were conducted in triplicate for each sample using Takyon Low ROX SYBR 2X MasterMix blue dTTP (Eurogentec, Cat. No. UF-LSMT-B0701, Fremont, CA, USA) and KiCqStart^®^ SYBR^®^ Green pre-designed primers (Appendix A) on a QuantStudio 6 Flex Real-Time PCR system (Thermo Fisher Scientific, Cat. No. 4485699). GAPDH gene served as a sample control (Appendix A). The comparative Ct method (or ΔCt) was used to analyze the data. 

### 2.9. Statistical Analysis

BUB1 expression data were analyzed using unpaired *t*-tests and ordinary one-way ANOVA. Survival analyses were conducted using the Kaplan–Meier method. Cell proliferation data were analyzed using a nonlinear regression curve fit. The Combination Index (C.I.) for double drug combinations was determined using Compusyn software, Version 1.0, and the Chow-Talay C.I. calculation formula: C.I. = (D)1/(Dχ)1 + (D)2/(Dχ)2, where (Dχ)1 and (Dχ)2 represent the concentrations of each drug alone to achieve χ% effect, while (D)1 and (D)2 denote the concentrations of drugs in combination to produce the same effect. A C.I. value less than 1, equal to 1, and greater than 1 indicates synergism, additivity, and antagonism, respectively [23]. For triple drug combinations and double drug combinations with radiation, the Bliss independence model was used to determine synergism, additivity, and antagonism. The model calculates the expected combined effect, E_AB_ of two drugs under the assumption of no interaction using the formula, E_AB_ = E_A_ + E_B_ (1 − E_A_), where E_A_ and E_B_ represent the observed effects of drugs A and B, respectively. Drug combinations are considered synergistic if the observed effects are higher than expected, antagonistic if lower, and additive if equal [24,25]. qPCR data were analyzed by ordinary one-way ANOVA with Dunnett’s multiple comparisons test. Graphs were generated on GraphPad Prism v8. Results are expressed as the mean ± standard error of the mean (SEM). Statistical significance was defined as *p*-values of 0.05 or less. Each experiment was performed in triplicate and repeated at least three times.

## 3. Results

### 3.1. BUB1 Is Overexpressed in LUAD, LUSC, and SCLC, and Is Correlated with Poorer Survival

BUB1 is significantly overexpressed in LUAD (n = 517, *p* = 7.4 × 10^−40^; Figure 1A) and LUSC (n = 501, *p* = 8.1 × 10^−34^; Figure 1B) compared to normal tissues (n = 59, 51) in the TCGA dataset. We also observed higher BUB1 expression in LUSC (n = 100, *p* = 1 × 10^−17^) as compared to LUAD (n = 85; Figure 1C) in the Rousseaux dataset. Surprisingly, significantly higher BUB1 was observed in SCLC (n = 21, *p* = 1.5 × 10^−10^; Figure 1D) as compared to LUAD in the Rousseaux dataset. Higher BUB1 expression correlated with poorer OS in LUAD (*p* = 0.0033; Figure 1E). Importantly, higher BUB1 expression was observed in TP53 mutated LUAD (n = 233) and LUSC (n = 369) compared to TP53 non-mutant cases (UALCAN TCGA dataset; Figure 1F,G), suggesting a potential association between p53 mutation and BUB1 expression. 

### 3.2. BUB1 Protein Is Overexpressed in Lung Tumor Tissues

Immunostaining in lung TMAs demonstrated that BUB1 was overexpressed in tumors (n = 274) compared to normal tissues (n = 34; Figure 1H). Within tumors BUB1 expression was higher in NSCLC (n = 206) as compared to SCLC (n = 68; Figure 1I). However, there was no correlation between BUB1 expression and tumor grade (Figure 1J) or stage (Figure 1K). Representative images showing differential BUB1 immunoreactivity (Figure 1L). BUB1 is overexpressed in NSCLC (NCI-H1975, A549, NCI-H2030) and SCLC (NCI-H2198, NCI-H1876) cell lines as compared to normal lung epithelial PCS-300-010 cells (Figure 1M).

### 3.3. BUB1 Inhibition Suppresses Cell Proliferation and Clonogenic Capacity

BUB1 inhibitor BAY1816032 dose-dependently suppressed proliferation of lung cancer cells in the Alamar blue cytotoxicity assay. The IC_50_ in normal lung epithelial PCS-300-010 cells was 15.9 μM (Figure 2A), while it was significantly lower (IC_50_ 1.1–5.1 μM) in lung cancer cell lines (Figure 2B–G). The IC_50_ for single-agent BUB1i dropped to the nano-molar range in clonogenic survival assays in A549, H2030, H1975, and Calu-1 cells (<250–900 nM; Figure 2H–K) indicating that clonogenic assays are more sensitive in finding dose effects on cytotoxicity. The SCLC cell lines did not form well-differentiated colonies; thus, the effect of BAY1816032 in these cells could not be evaluated by clonogenic survival assay. To further confirm that the effect of BAY1816032 was indeed due to the presence of BUB1 protein, we next depleted endogenous BUB1 by siRNA and treated these cells with BAY1816032. Not surprisingly, there was no effect of BAY1816032 on the growth of BUB1-depleted A549 and H1975 cells (Appendix A).

### 3.4. The Sequence and Doses of BUB1 Inhibitor Determine Sensitivity to Paclitaxel, Cisplatin, and Olaparib

Normal bronchial/tracheal epithelial NSCLC and SCLC cell lines displayed varying toxicities to single-agent paclitaxel, cisplatin, and olaparib with IC_50_ values ranging from 4 to 44 μM for cisplatin (Figure 3A–G), 2.5 to 18.2 nM for paclitaxel (Figure 3H–N), and 2.4 to 48.7 μM for olaparib (Figure 3O–U and Appendix A). Although the single agent IC_50_ in PCS-300-010 cells was observed in the higher concentration range (Figure 3A,H,O), the IC_50_ in lung cancer cell lines were found to be higher (NCI-H1876 44 µM cisplatin; H2030 18.2 nM paclitaxel; H2030 48.7 µM olaparib). 

The NCI-H1975 cell line was chosen to find the optimal dosing schedule for combination treatments since these cells demonstrated moderate sensitivity to single-agent treatments in cytotoxicity assays. The single agent IC_50_ for cisplatin, paclitaxel, and olaparib was reduced by a log concentration (Figure 4A–C) compared to the cytotoxicity assay (Figure 3D,K,R). Pretreatment with BUB1i 24 h prior to cisplatin resulted in antagonism (C.I. = 1.22 Figure 4D) which was further potentiated at a higher cisplatin concentration (C.I. = 1.27; Figure 4D). Similarly, pretreatment with cisplatin followed by BUB1i exhibited antagonism (C.I. = 1.5, Figure 4E). The antagonism slightly reduced when a higher BUB1i was used while maintaining constant cisplatin (C.I. = 1.1; Figure 4E). Concurrent BUB1i and cisplatin led to synergism (C.I. = 0.9; Figure 4F) which shifted to additivity with a higher cisplatin concentration with a constant BUB1i (C.I. = 0.9 to 1.1; Figure 4F). Conversely, increasing the BUB1i dose with constant cisplatin switched antagonism to additivity (C.I. = 1.5 to 1.1; Figure 4G). 

BUB1i and paclitaxel sequential treatments in either schedule reduced colony-forming abilities additively (C.I. > 1; Figure 4H,I). BUB1i concurrently with paclitaxel led to synergistic sensitization (C.I. < 1; Figure 4J,K). Increasing concentrations of paclitaxel (Figure 4J) or BUB1i (Figure 4K) demonstrated more potent synergism. BUB1i prior to olaparib was additive (Figure 4L) while olaparib pretreatment was synergistic (Figure 4M). Concurrent BUB1i and olaparib administration resulted in such low survival fractions that C.I. could not be calculated (Figure 4N,O). Based on the above results (Figure 4D–O), concurrent BUB1i with other agents was selected as the preferred approach in subsequent experiments. 

### 3.5. BUB1 Inhibition Chemosensitizes NSCLC in Double and Triple Drug Combinations

Concurrent BUB1i with cisplatin is antagonistic in A549 and H2030 cells (Figure 5A,B) with additivity observed at higher concentrations in A549 (Figure 5A) which is similar to H1975 (Figure 4F,G). Concurrent BUB1i demonstrated potent synergism with paclitaxel in A549 and H2030 cells (C.I. < 1; Figure 5C,D). BUB1i synergistically sensitizes A549 cells to olaparib (Figure 5E) while it demonstrates synergism or additivity in H2030 cells depending on the doses (Figure 5F). 

Multimodal therapeutic combinations may provide better benefits in aggressive and recurrent cancers [3,26]. Several clinical trials such as trials on NCT04624204, NCT05298423, and NCT03467360 are evaluating the effectiveness of multidrug combinations including targeted therapy in lung cancers; we next tested if combining BUB1i with paclitaxel and olaparib (trimodal therapy) would further enhance cytotoxicity. The triple drug combination of BAY1816032, paclitaxel, and olaparib was most cytotoxic in A549, H2030, and H1975 (Figure 5G–I), suggesting potential therapeutic benefits of including BUB1i in NSCLC management. The Bliss Independence Model was used to evaluate the effects of drug combinations on cell survival fractions (SF). According to this model, synergism occurs when the observed effect (OE) of combining drugs is higher than the expected effect (EE) calculated from the individual effects of each drug [24]. This means the combination is synergistic if the observed survival fraction (*SFobs*.) is lower than the expected survival fraction (*SFexp*.) indicating that the drugs work better together than independently. Specifically, in A549 cells, the *SFexp*. was 0.99 and *SFobs*. was 0.05 (Figure 5G), in H2030 cells, the *SFexp*. was 0.98 and *SFobs*. was 0.25 (Figure 5H), while in H1975 cells, the *SFexp*. was 1 and *SFobs*. was 0.36 (Figure 5I).

### 3.6. BUB1 Inhibition Sensitizes NSCLC to Radiation

BAY1816032 dose-dependently increased radiation-induced cell death in A549, H2030, and H1975 cells (rER >1; Figure 6A–C). The *SFobs*. of cells treated with increasing doses of BUB1i and radiation was lower than the *SFexp.* calculated from the individual effects of BUB1i or radiation indicating a synergistic effect. In Calu-1 cells, the IC_50_ for BAY1816032 monotherapy was 2.8 µM (Figure 2E), which decreased to 1.9 µM when combined with radiation (4Gy; Appendix A). Similarly, the IC_50_ for cisplatin, paclitaxel, and olaparib alone also decreased from 4.8 µM, 13.2 nM, and 22.1 µM (Figure 3E,L,S) to 2.1 µM, 5.1 nM, and 5.6 µM, respectively, when combined with radiation (4Gy; Appendix A). The IC_50_ for BUB1i in normal lung epithelial PCS-300-010 cells was significantly higher compared to cancer cell lines (15.9 µM; Figure 2A), with only a slight decrease (14 µM) when combined with IR (Appendix A), indicating selectivity. In the presence of IR, the IC_50_ for cisplatin increased from 35 µM to 44 µM while the IC_50_ for paclitaxel and olaparib decreased only slightly in PCS-300-010 cells (Figure 3A,H,O and Appendix A). These results signify a role for BUB1i in radiosensitizing NSCLC while potentially minimizing toxicity in normal tissues.

### 3.7. BUB1 Inhibition Sensitizes NSCLC and SCLC to Chemoradiation

BUB1 inhibitor BAY1816032 increases radiosensitization by cisplatin (BAY1816032 + cisplatin + IR; Figure 6D–F), paclitaxel (BAY1816032 + paclitaxel + IR; Figure 6G–I), and olaparib (BAY1816032 + olaparib + IR; Figure 6J–L) in NSCLC (A549, H2030 and H1975) cell lines. This potential synergistic sensitization increased with increasing radiation doses across all tested combinations (Figure 6D–L). The plating efficiency (PE) and SFs at 2 Gy are shown in the inset in Figure 6A–C. 

The response of SCLC to BUB1i was variable in double drug or radiation combinations. BUB1i was antagonistic with cisplatin in the NCI-H1876 cell line (Figure 7A; black bars) which shifted to synergism/additivity when combined with IR (Figure 7A; magenta bars). BUB1i synergistically sensitized NCI-H2198 cells to cisplatin which was independent of radiotherapy (Figure 7B). Similarly, BUB1i and paclitaxel combination was synergistic in NCI-H1876 and NCI-H2198 cells independent of presence or absence of IR (Figure 7C,D). BAY1816032 combination with olaparib was also synergistic in both the SCLC cell lines even in the absence of IR (Figure 7E,F).

### 3.8. BUB1 Inhibition Delays DNA Double-Strand Break Repair and Elicits Pro-Apoptotic and Anti-Proliferative Responses

DNA double-strand breaks (DSBs) were evaluated by γ-H2AX foci assay in H2030 cells since these cells form clear γ-H2AX foci in response to IR. BUB1 inhibition led to the persistence of γ-H2AX foci 24 h post-irradiation (Figure 8A,B) while most foci were repaired in IR-only samples by 24 h, indicating that the DNA DSB repair pathway was impacted. Next, we evaluated if BUB1i and IR had any effect on apoptotic and proliferation markers. BUB1i increased BAX and decreased BCL2 and PCNA expression, which is indicative of a shift towards pro-apoptotic signaling and the downregulation of cellular proliferation (Figure 8C). Elevated expression of Caspase 9, Caspase 3, and TP53BP1 with BUB1i (Figure 8C) indicated enhanced apoptotic response post-treatment.

## 4. Discussion

Our observation that BUB1 is upregulated across various lung cancer subtypes (Figure 1A–D) and its overexpression correlates with survival is consistent with earlier reports [13,14,15,27] and reemphasizes its potential as a crucial molecular target in lung cancer. Furthermore, our observations of BUB1’s association with TP53 mutations in LUAD and LUSC (Figure 1F,G) support a role for p53 in potentially regulating BUB1 expression in lung cancer [28] and as a potential synthetic lethal combination with p53 [29]. Confirmation of BUB1 protein overexpression in lung tumor TMAs (Figure 1H,I,L) and lung cancer cell lines (Figure 1M) further emphasizes its significance as a potential biomarker and a molecular target in lung cancer. Surprisingly, we did not find any correlation between BUB1 expression and cancer stage or grade (Figure 1J,K), which is consistent with the literature [14,30]. However, this needs to be reconfirmed in a larger lung TMA sample set.

The BUB1i dose required to kill normal cells was ~4–10 times (Figure 2A) higher than that required in NSCLC and SCLC (Figure 2B–G) cells, further confirming the selectivity of BUB1i towards cancer cells. Clonogenic assays proved their higher sensitivity (than cytotoxic assays) and confirmed the lasting impact of BAY1816032 on reducing clonogenic capacity at much lower doses (Figure 2H–K). Similarly, the IC_50_ of cisplatin, paclitaxel, and olaparib in cytotoxicity assays (Figure 3A–U) was higher than in the colony formation assays (Figure 4A–C). Our systematic drug schedule experiments established that a concurrent combination of BUB1i with cisplatin, paclitaxel, and olaparib resulted in the highest cytotoxicity in lung cancer cells (Figure 4D–O). While BAY1816032’s efficacy in double drug combinations has been previously evaluated [19], this study for the first time confirmed that BAY1816032 was very effective at chemosensitizing lung cancer cell lines when combined with two different classes of approved therapies (Figure 5G–I). 

We envision complex signaling interplay in BUB1-mediated sensitization to cisplatin, paclitaxel, and olaparib and these agents together with radiation (Figure 9). We propose that the complex signaling interplay is based on distinct mechanisms of action of the drugs and BUB1’s role in mitotic regulation and DNA damage repair. Based on our observations that sequential treatment of BAY1816032 and cisplatin led to antagonism (Figure 4D,E), while concurrent treatment demonstrated synergism (Figure 4F,G), we hypothesize that antagonism during sequential treatment occurs because cisplatin induces DNA damage that arrests the cell cycle in the S and G2 phases [19], preventing cells from entering mitosis, where BUB1 is essential. Since BUB1 inhibition targets the spindle assembly checkpoint during mitosis, its effectiveness is diminished when the cell cycle is halted earlier by cisplatin. Similarly, if BUB1 inhibition is applied first, it may stall cells in a phase where cisplatin is less effective, leading to antagonism. Additionally, BAY1816032 may influence DNA damage response pathways, particularly non-homologous end joining (NHEJ), thereby reducing the efficacy of subsequent cisplatin treatment, and contributing to antagonism. In contrast, the synergy observed with concurrent treatment likely results from the simultaneous induction of DNA damage by cisplatin and mitotic errors by BUB1 inhibition, overwhelming the cells and leading to enhanced cell death. Concurrent administration may allow both agents to act on different but complementary cellular targets simultaneously, resulting in increased synergy. Thus, the timing and sequence of drug administration are crucial in determining the interaction between BAY1816032 and cisplatin. This hypothesis will be further explored in the future to optimize combination therapies involving BUB1 inhibitors. 

When BAY1816032 is combined with paclitaxel, the effects are additive in sequential treatment and synergistic in concurrent treatment. We predict that in sequential treatment, cells can partially address the spindle attachment errors caused by paclitaxel before BUB1 inhibition occurs, leading to a cumulative but not amplified effect. However, concurrent administration may overwhelm the cell by inducing spindle errors via paclitaxel while inhibiting their correction through BUB1 inhibition resulting in severe chromosomal instability and increased cell death leading to synergism. Similarly, sequential BAY1816032 with olaparib was additive and the concurrent treatment was synergistic. We envision that the sequential treatment may allow cells to partially restore DNA repair before BUB1 inhibition disrupts the mitotic checkpoint. We envision that concurrent administration may simultaneously impose stress on both DNA repair mechanisms and mitotic processes leading to higher chromosomal instability and cell death, resulting in potent synergism.

BUB1 has been recognized as a potential target for radiosensitization [31], and our group and others have reported that it plays an integral role in DNA damage response [11,20,32,33,34]. In this comprehensive study, we not only establish that BUB1i radiosensitizes NSCLC (Figure 6A–C) but also demonstrate that BUB1i radiosensitizes NSCLC in combination with cisplatin, paclitaxel, and olaparib (Figure 6D–L). Additionally, this is the first study that demonstrates that BUB1 inhibitor chemo- and radiosensitizes SCLC cell lines (Figure 7A–F). The persistence of γ-H2AX foci in H2030 upon BUB1 inhibition (Figure 8A,B) indicative of a delayed DSB repair confirms a role for BUB1 kinase activity in this process. Furthermore, an increase in 53BP1 in response to BAY1816032 and IR suggests a role for BUB1 in an active DNA damage response [35]. Altered expression of key markers such as BAX, BCL2, PCNA, and Caspases 9 and 3 (Figure 8C) after BUB1i + IR indicates a shift toward a pro-apoptotic and anti-proliferative state. Similar pro-apoptotic responses have been reported by our group and others in response to BUB1 inhibition [18,20,34,36] that emphasize a potential commonality in the molecular responses triggered by BUB1 across different cancers.

PARP1/2 proteins usually detect SSB and recruit factors to repair the SSB. PARPi causes either PARP trapping on DNA break sites that leads to replication fork collapse and cell death, especially in BRCA mutant cell lines [37], or PARPi can upregulate NHEJ and reduce HR leading to genomic instability and cell killing [38]. We hypothesize that BUB1i sensitizes to PARP inhibitors because PARPi can increase the dependency on NHEJ for which BUB1 is critical [20]. Platinums (cisplatin, carboplatin) form DNA adducts that cause DNA replication errors leading from SSB to DSB, cell-cycle arrest in G1-S, and ultimately cell death [39,40]. Increased platinum-DNA adduct repair has been shown to be associated with cisplatin resistance [41]. Similarly, paclitaxel blocks the depolarization of microtubules, leading to improper chromosome segregation, and G2/M cell-cycle arrest [42,43] resulting in apoptotic cell death [44,45]. NSCLC cell lines have variable sensitivity to these drugs. BUB1 not only regulates the cell cycle but also regulates DNA damage response. Therefore, we propose that BUB1 inhibition sensitizes these agents and overcomes resistance because of its ability to target multiple pathways. In summary, our data demonstrate that BUB1i increases NSCLC and SCLC cell death by paclitaxel, cisplatin, and olaparib and that these effects are accentuated when combined with radiation. Importantly, the ability of BUB1i to sensitize BRCA efficient cell lines to PARPi (olaparib) opens a new horizon for treating/sensitizing these tumors with anti-PARP agents. The sensitization of NSCLC and SCLC by BUB1i, cisplatin, paclitaxel, olaparib, and IR suggest complex interactions involving DNA damage repair pathways and modulation of apoptotic signaling. Future studies will decipher these complex mechanisms. Since BUB1 expression directly correlates with p53 (TP53; [46]), we will investigate a potentially synthetic lethal relationship between BUB1 inhibition and TP53 mutations that could help identify biomarkers that predict sensitivity to BUB1 inhibitors in TP53-mutant cancers including in lung cancers.

Our current study was limited to non-small cell lung cancer (NSCLC) and small cell lung cancer (SCLC) cell lines. In the future, we will expand these observations by including squamous cell carcinoma (e.g., HCC95, LK-2, EBC-1) and large cell lung carcinoma (e.g., LU65, LU99) cell lines. Additionally, we will translate our in vitro findings to tumor xenografts including the syngeneic mouse model. These approaches will further strengthen the therapeutic targeting of BUB1 in lung cancer. Furthermore, we will delineate the molecular mechanisms through which BUB1-mediated sensitization takes place. This will likely involve DNA damage repair pathways, apoptotic signaling cascades, and cell-cycle signaling. Developing biomarkers to predict the response to BUB1 inhibition would enable more personalized and effective treatment approaches. These future studies will be critical for fully realizing BUB1’s potential as a therapeutic target in lung cancer. In the future, we anticipate a phase-1 clinical trial that evaluates the safety of BUB1 inhibitor BAY1816032 as a monotherapy and in combination with cisplatin, paclitaxel, PARP inhibitors, and radiotherapy in patients with lung cancer. 

## 5. Conclusions

In this study, we show a direct correlation between BUB1 protein expression and overall survival and establish BUB1 as a novel molecular target for enhancing the therapeutic potential of cisplatin, paclitaxel, olaparib, and ionizing radiation in lung cancer. Our findings highlight the potential of BUB1 inhibition as a promising approach to augment the effectiveness of radiotherapy and chemoradiation in NSCLC and SCLC.

## Figures and Tables

**Figure 1 cancers-16-03291-f001:**
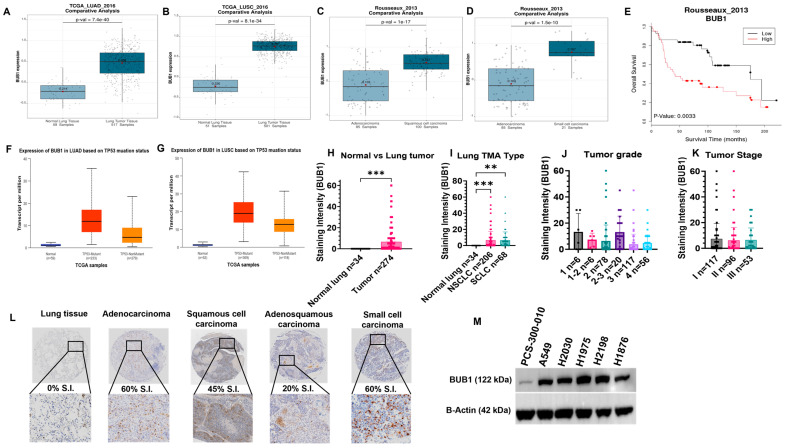
BUB1 is overexpressed in lung cancers and its overexpression correlates with poorer survival. (**A**) BUB1 is overexpressed in LUAD (n = 517, *p* = 7.4 × 10^−40^) compared to normal tissues (n = 59) in the Lung explorer database, TCGA_LUAD_2016 dataset. (**B**) BUB1 is overexpressed in LUSC (n = 501, *p* = 8.1 × 10^−34^) compared to normal tissues (n = 51) in the TCGA_LUSC_2016 dataset. (**C**,**D**) BUB1 is expressed higher in LUSC (n = 100, *p* = 1 × 10^−17^) and SCLC (n = 21, *p* = 1.5 × 10^−10^) than LUAD (n = 85) in the Lung explorer database, Rousseaux_2013 dataset. (**E**) Higher BUB1 expression significantly correlates with poorer survival in LUAD (*p* = 0.0033) in the Rousseaux_2013 dataset. (**F**,**G**) BUB1 expression correlates with TP53 mutation status in LUAD (n = 233) and LUSC (n = 369) in the TCGA dataset in the UALCAN database. (**H**) BUB1 is significantly overexpressed in lung tumors compared to normal tissues. (**I**) NSCLC (n = 206), and SCLC (n = 68) express significantly higher BUB1 than normal tissues (n = 34). (**J**,**K**) Correlation between BUB1 expression and tumor grade and stage in the TMAs. (**L**) BUB1 expression intensity in representative lung cancer subtypes. (**M**) BUB1 expression in lung cancer cell lines and normal bronchial epithelial cells. ** *p*  ≤  0.01; *** *p*  ≤  0.001. The original Western blot figures can be found in Appendix A.

**Figure 2 cancers-16-03291-f002:**
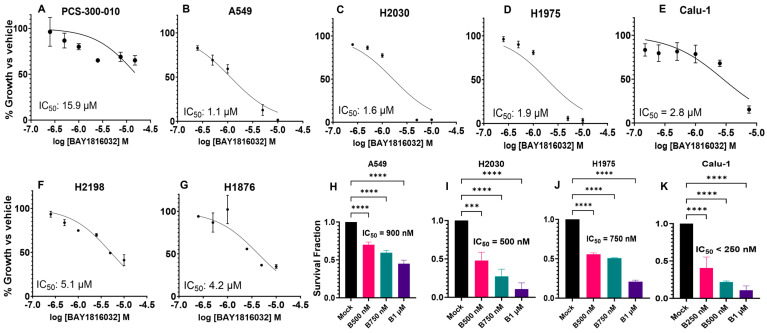
BUB1 inhibitor is cytotoxic in NSCLC and SCLC cell lines. Alamar blue cell proliferation assay in (**A**) normal lung PCS-300-010, and lung cancer cell lines, (**B**) A549, (**C**) H2030, (**D**) H1975, (**E**) Calu-1, (**F**) H2198, and (**G**) H1876. Clonogenic survival assay in (**H**) A549, (**I**) H2030, (**J**) H1975, and (**K**) Calu-1 with different BUB1i concentrations. *** *p*  ≤  0.001; **** *p*  ≤  0.0001.

**Figure 3 cancers-16-03291-f003:**
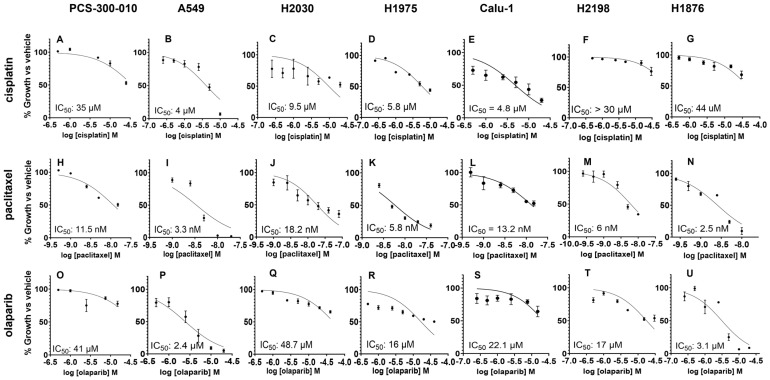
Cytotoxicity of single-agent cisplatin, paclitaxel, and olaparib in normal lung and NSCLC and SCLC cell lines and estimation of inhibitory concentration 50% (IC_50_). The effect of cisplatin (**A**–**G**), paclitaxel (**H**–**N**), and olaparib (**O**–**U**) on cell growth was measured by Alamar blue assay in bronchial epithelial (PCS-300-010), NSCLC (A549, NCI-H2030, NCI-H1975, Calu-1), and SCLC (NCI-H2198, NCI-H1876) cells. Each column represents a cell line.

**Figure 4 cancers-16-03291-f004:**
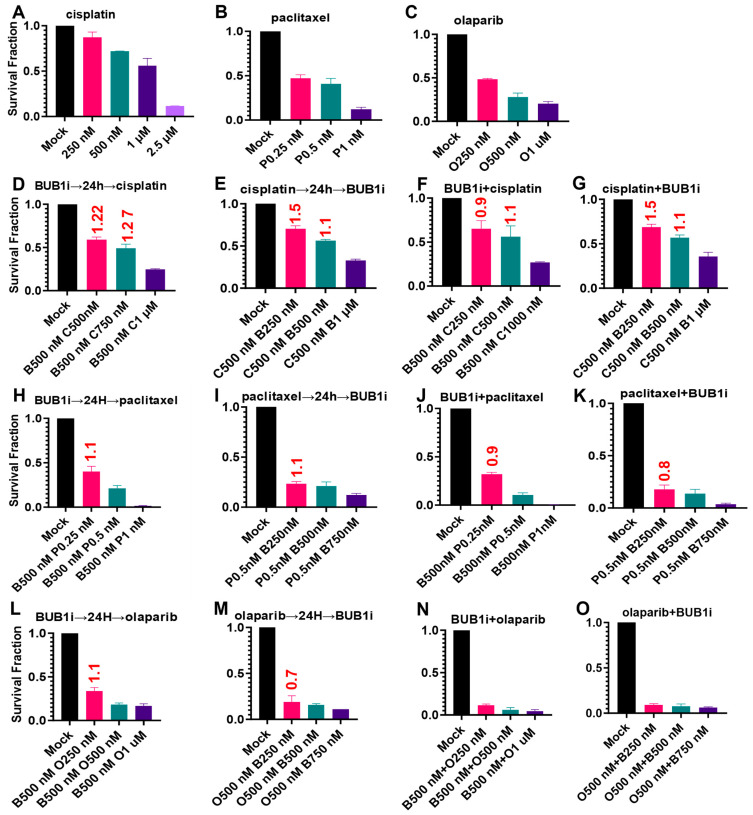
Concurrent treatment with BUB1 inhibitor BAY1816032 sensitizes the NCI-H1975 NSCLC cell line to cisplatin, paclitaxel, and olaparib. (**A**–**C**) The IC_50_ of single agent cisplatin, paclitaxel, and olaparib was estimated in colony formation assays. (**D**,**E**) Sequential treatment of BUB1i with cisplatin, paclitaxel (**H**,**I**), olaparib (**L**,**M**), and concurrent (**F**,**G**,**J**,**K**,**N**,**O**) for identifying the optimal dosing schedule. The combination index (C.I.) was estimated as described in Section 2. C.I. > 1, =1 and <1 indicate antagonism, additivity, and synergism, respectively.

**Figure 5 cancers-16-03291-f005:**
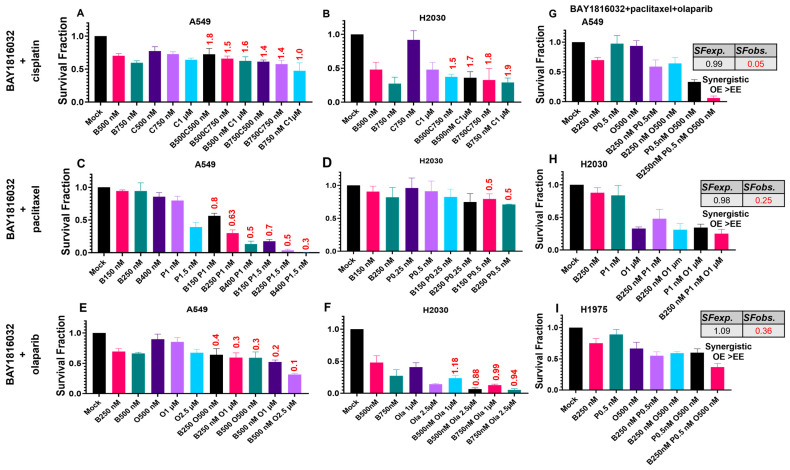
BUB1i sensitizes NSCLC to cisplatin, paclitaxel, and olaparib in double and triple drug combinations. (**A**) BUB1 inhibitor sensitized A549 cells to cisplatin additively at higher concentrations (BAY1816032 750 nM and cisplatin 1 µM; C.I. = 1.0). (**B**) BUB1i was antagonistic with cisplatin in H2030 cells at all the concentrations tested. (**C**,**D**) BAY1816032 synergistically enhanced cell killing when combined with paclitaxel in NSCLC. (**E**,**F**) BAY1816032 demonstrated synergism with olaparib in A549 while it was synergistic or additive in H2030 depending on the doses. (**G**–**I**) The triple drug combination of BAY1816032 with paclitaxel and olaparib had higher cytotoxicity even when used at lower concentrations than the drug concentrations used in double combinations. (**G**) In A549 cells, the *SFexp*. was 0.99 and *SFobs*. was 0.05; in H2030 cells, the *SFexp*. was 0.98 and *SFobs*. was 0.25, while in H1975 cells, the *SFexp*. was 1 and *SFobs*. was 0.36.

**Figure 6 cancers-16-03291-f006:**
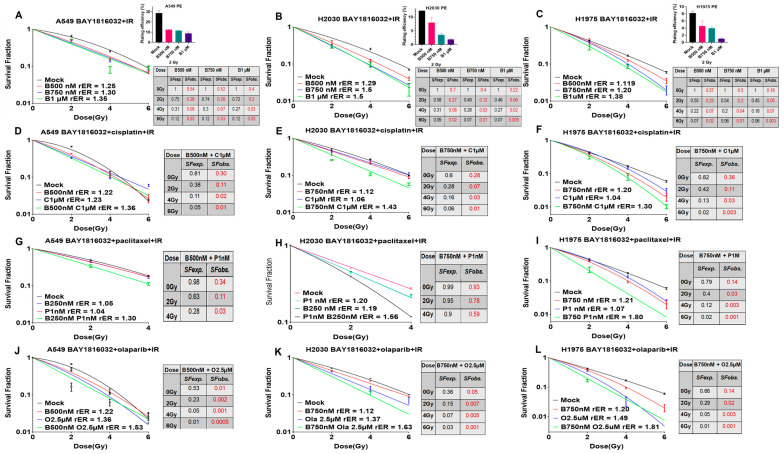
BUB1 inhibition sensitizes NSCLC cell lines to radiotherapy and chemoradiation. (**A**–**C**) BAY1816032 radiosensitizes A549, H2030, and H1975 cell lines at drug concentrations below the IC_50_. The radiation enhancement ratio (rER) ranges from 1.35 to 1.5 at 1µM BUB1i concentration. (**D**–**F**) BUB1i significantly increased the radiosensitization potential of cisplatin, paclitaxel (**G**–**I**), and olaparib (**J**–**L**) in NSCLC cell lines. Inset tables show the survival fractions of BUB1i with chemotherapy and radiotherapy.

**Figure 7 cancers-16-03291-f007:**
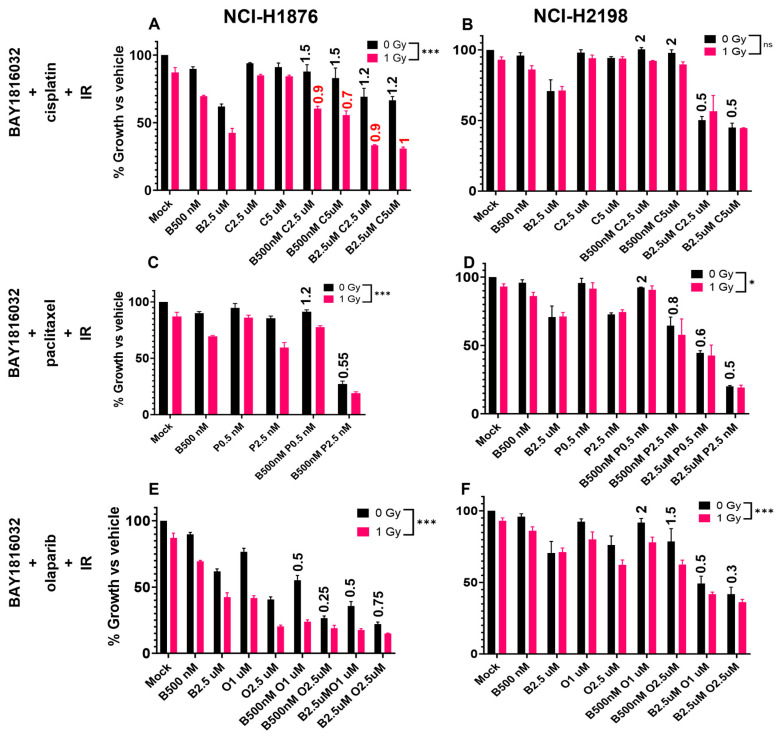
BUB1i sensitizes SCLC to cisplatin, paclitaxel, olaparib, and radiation in multimodal therapy combinations. (**A**) BAY1816032 is antagonistic with cisplatin in the NCI-H1876 cell line (black bars) which is shifted to synergism/additivity when combined with radiation (magenta bars). (**B**) BAY181062 demonstrated synergistic sensitization with cisplatin in NCI-H2198 cells independent of radiotherapy. (**C**,**D**) BUB1i demonstrated synergism with paclitaxel in NCI-H1876 and NCI-H2198 cell lines independent of radiation. (**E**,**F**) BAY1816032 synergistically sensitized both the SCLC cell lines to olaparib independent of IR. C.I. values are shown in black for no IR treatment, and in red for IR treatment. * *p*  ≤  0.05; *** *p*  ≤  0.001. ns not significant.

**Figure 8 cancers-16-03291-f008:**
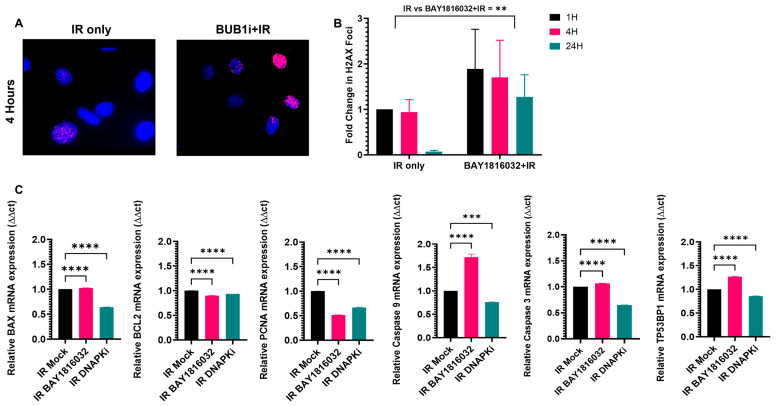
BUB1 inhibition delays radiation-induced DNA DSB repair and promotes apoptosis. (**A**) BUB1i delayed γH2AX foci resolution in the H2030 cell line. The nucleus is stained blue with DAPI, while γH2AX foci are stained red. (**B**) BUB1i caused a statistically significant increase in radiation-induced γH2AX foci which persisted long-term. (**C**) BUB1i with IR significantly increased pro-apoptotic markers and decreased anti-proliferative markers in qRT PCR assays in an NSCLC cell line. ** *p*  ≤  0.01; *** *p*  ≤  0.001; **** *p*  ≤  0.0001.

**Figure 9 cancers-16-03291-f009:**
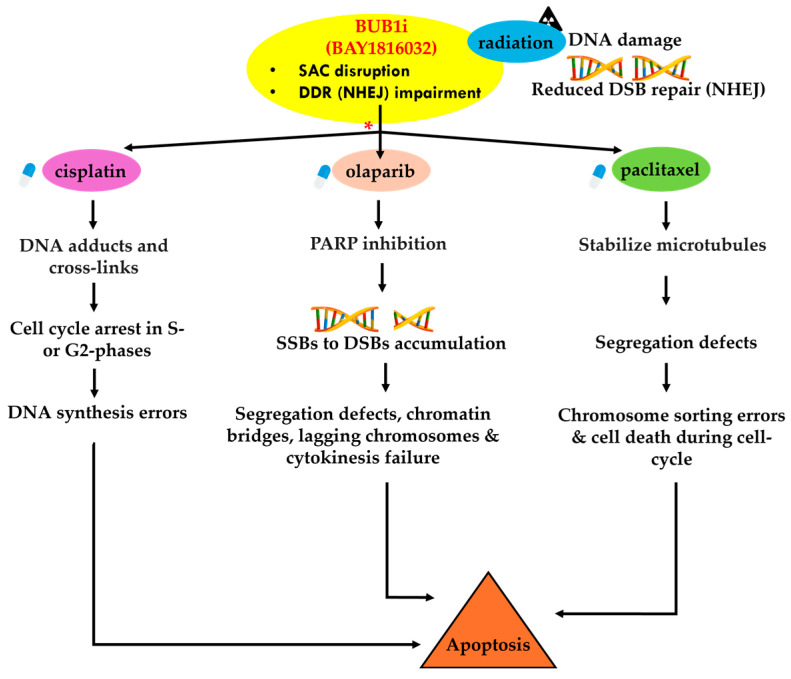
* Proposed mechanism by which BUB1 inhibition (BAY1816032) sensitizes lung cancer cells to platinum (cisplatin), microtubule inhibitor (paclitaxel), and PARP inhibitor (olaparib) and radiotherapy. We propose that BUB1 inhibitor BAY1816032 disrupts the spindle assembly checkpoint (SAC) which is a key regulator of mitosis and impairs radiation-induced DNA double-strand breaks repair, particularly NHEJ. This disruption leads to the accumulation of DNA damage and radiation sensitization. The effects of BUB1i and radiotherapy on cell-cycle arrest and DNA DSB repair are compounded when combined with drugs that form DNA adducts (cisplatin), stabilize microtubules (paclitaxel), or inhibit DNA repair (olaparib) leading to increased cell death.

## Data Availability

The data are available from the corresponding author(s) on reasonable request.

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
