# Peer review of "BUB1 Inhibition Overcomes Radio- and Chemoradiation Resistance in Lung Cancer"

_cancers, 2024, doi:10.3390/cancers16193291_

Round 1

Reviewer 1 Report

Comments and Suggestions for Authors

Shivani and co-workers presented an article titled, “BUB1 inhibition overcomes radio- and chemoradiation resistance in lung cancer”. The authors aimed to demonstrate the impact of BUB1 inhibitors on cancer therapeutic outcomes. The authors employed various biochemical and molecular assays including immunostaining of lung tumor microarrays. Authors identified that inhibition of BUB1 sensitizes NSCLC and SCLC to chemotherapies (cisplatin, paclitaxel), targeted therapy (olaparib), and radiation therapy. Further, the authors concluded that BUB1 inhibitors also sensitized NSCLC and SCLC to radiotherapy and chemoradiation suggesting the concept can be a new approach to cancer therapy. The presented study is an extension of the author's previously published papers.

The work was presented clearly and the literature covers all the basic information about the study in the introduction section.

References covered recent literature and are complete/adequate.

The manuscript discussed the obtained results and the conclusion part is in alignment with the aim of this study.

All the figures are appropriately presented, clearly.

However, the below-mentioned minor revision is required before the acceptance of the manuscript:-

1.     Figure 1E is to be presented clearly with high-resolution

2.     Authors mentioned Rousseaux_2016 datasets from LCE in the materials section, however, in figure 1, it was indicated as Rousseaux_2013 dataset. Authors should address this for more clarity. Further, the dataset is more than a decade, why recent years datasets were not considered in the study.

3.     Authors can propose a mechanism for the events observed during the study as a Figure 9 and discuss it in more detail, accordingly.

4.     In conclusion, authors may indicate the future of the work.

Author Response

Reviewer 1:

 Comments and Suggestions for Authors

Shivani and co-workers presented an article titled, “BUB1 inhibition overcomes radio- and chemoradiation resistance in lung cancer”. The authors aimed to demonstrate the impact of BUB1 inhibitors on cancer therapeutic outcomes. The authors employed various biochemical and molecular assays including immunostaining of lung tumor microarrays. Authors identified that inhibition of BUB1 sensitizes NSCLC and SCLC to chemotherapies (cisplatin, paclitaxel), targeted therapy (olaparib), and radiation therapy. Further, the authors concluded that BUB1 inhibitors also sensitized NSCLC and SCLC to radiotherapy and chemoradiation suggesting the concept can be a new approach to cancer therapy. The presented study is an extension of the author's previously published papers.

The work was presented clearly, and the literature covers all the basic information about the study in the introduction section. References covered recent literature and are complete/adequate. The manuscript discussed the obtained results, and the conclusion part is in alignment with the aim of this study.

All the figures are appropriately presented clearly.

However, the below-mentioned minor revision is required before the acceptance of the manuscript:-

We thank the reviewer #1 for their positive feedback on our work. Our response to minor critiques is provided below.

  1. Figure 1E is to be presented clearly with high resolution.

Thank you for your feedback. We recognize the importance of providing high-resolution images for clarity. The original Figure 1E was sourced directly from the LCE website, which restricted our ability to edit it. However, we have now enlarged the figure to improve its resolution and ensure better visibility in the revised manuscript.

  1. Authors mentioned Rousseaux_2016 datasets from LCE in the materials section, however, in figure 1, it was indicated as Rousseaux_2013 dataset. Authors should address this for more clarity. Further, the dataset is more than a decade, why recent years datasets were not considered in the study.

We appreciate the reviewer for highlighting the error. We have corrected it from Rousseaux_2016 to Rousseaux_2013 in the materials section for accuracy. Please refer to page 2, line 80. Regarding the use of older datasets, while newer ones are available, we chose the Rousseaux_2013 dataset from the publicly accessible LCE database. This dataset was selected because it includes samples from both SCLC and NSCLC subtypes (LUAD and LUSC), allowing for a more comprehensive comparison across these cancer types. Other databases such as UALCAN (TCGA), do not include SCLC samples, which would have limited the scope of our analysis.

  1. Authors can propose a mechanism for the events observed during the study as a Figure 9 and discuss it in more detail, accordingly.

Thank you for the valuable suggestion. We appreciate your input, which has greatly contributed to improving the clarity and depth of our study. We have now included a “potential mechanism” through which BUB1 may sensitize lung cancer to chemotherapy (cisplatin, paclitaxel, olaparib) and radiation. In this mechanism, we propose that BUB1 inhibitor BAY1816032 disrupts the proper functioning of the spindle assembly checkpoint (SAC), a key regulator of mitosis, thereby impairing DNA repair processes, particularly the non-homologous end joining (NHEJ) pathway of DNA damage repair. This disruption leads to the accumulation of DNA damage within cancer cells. When combined with chemotherapy and radiation, this overwhelms the DNA repair mechanisms, resulting in cell cycle checkpoint failure and mitotic catastrophe. The resulting chromosomal instability triggers the intrinsic apoptotic pathway, ultimately enhancing cancer cell death. We have added a detailed discussion of this mechanism, along with a figure, in the revised manuscript to offer a clearer understanding of how BUB1 inhibition interacts with chemotherapy and radiation treatments. Please refer to pages 13-14, lines 392-436, and figure 9.

  1. In conclusion, authors may indicate the future of the work.

Thank you for your suggestion. We have added a new section in the discussion outlining the future directions of our research. Please refer to page 15, lines 473-491.  In the current study, we utilized four NSCLC and two SCLC lung cancer cell lines. In the future, we will extend these findings to additional lung cancer cell lines including squamous cell carcinoma (HCC95, LK-2, EBC-1) and large cell lung carcinoma (LU65, LU99).

We intend to investigate the effects of BUB1 inhibition with cisplatin, paclitaxel, and olaparib (double combinations) and with radiation to delineate the underlying mechanism of sensitization. Additionally, we plan to conduct tumor xenograft studies in NOD-SCID mice (or potentially using the Lewis lung carcinoma syngeneic model in C57BL/6 mice) to validate the synergistic effects in these preclinical in-vivo models.

Additionally, we will explore the synthetic lethal relationship between BUB1 inhibition and TP53 mutations to investigate the mechanistic link between p53 and BUB1. These studies will help advance our understanding of the potential clinical applications of BUB1 inhibition in lung cancer.

Reviewer 2 Report

Comments and Suggestions for Authors

1) Please authenticate the cell lines by STR analysis.

2) NSCLC is a heterogeneous group of five different types of cancers. The three predominant types of cancers are adenocarcinomas, squamous cell carcinomas and large cell carcinomas. The authors have only used adenocarcinomas and squamous cell lung carcinomas. They should validate their results in a large cell carcinoma cell line.

3) The authors need to show that the growth suppressive activity of BAY1816032 in the lung cancer cell lines is indeed mediated by BUB1Kinase. A simple way to do this would to deplete BUB1 kinase by siRNA methodology and show that the BAY1816032 does not display growth-inhibitory activity in human lung cancer cell lines when BUB1 kinase is silenced.

4) A caveat of the paper is that it does not provide any insight into the signaling pathways underlying the growth-suppressive activity of BAY1816032, cisplatin and radiation. They have shown that BUB1 inhibition DNS strand break repair and promotes programmed cell death. They need to show the cytoplasmic signaling network at which these there converge and how that phenomena causes enhanced growth-suppressive activity in lung cancer cells.

5) The PI observed that "pretreatment with BAY1816032 24 hours prior to cisplatin resulted in antagonism, which potentiated at higher cisplatin concentration. Similarly, pretreatment with cisplatin followed by BAY1816032 exhibited antagonism. Concurrent BAY1816032 and cisplatin led to synergism". The authors should comment on these observations in the DISCUSSION and provide a possible hypothesis why the combination of BAY1816032 and cisplatin sometimes led to antagonistic interactions and sometimes led to synergy.

Author Response

Reviewer 2:

Comments and Suggestions for Authors

We thank the reviewer #2 for their thorough evaluation of our manuscript and helpful suggestions that resulted in significant improvement.

1) Please authenticate the cell lines by STR analysis.

We thank the reviewer for the suggestion. All the cell lines used in the current study were purchased from ATCC who have originally validated these cells. We plan to conduct STR analysis on these cells in the near future as we seek to get extramural funding to support the continuation of these studies.

2) NSCLC is a heterogeneous group of five different types of cancers. The three predominant types of cancers are adenocarcinomas, squamous cell carcinomas and large cell carcinomas. The authors have only used adenocarcinomas and squamous cell lung carcinomas. They should validate their results in a large cell carcinoma cell line.

Thank you for the feedback. We plan to incorporate additional NSCLC subtype cell lines such as large cell carcinoma as we expand our studies in the near future (see the response to Reviewer #1, comment #4).

3) The authors need to show that the growth suppressive activity of BAY1816032 in the lung cancer cell lines is indeed mediated by BUB1 kinase. A simple way to do this would deplete BUB1 kinase by siRNA methodology and show that the BAY1816032 does not display growth-inhibitory activity in human lung cancer cell lines when BUB1 kinase is silenced.  

We thank the reviewer for suggesting this important experiment. We conducted the experiment as advised by the reviewer in the A549 and H1975 cell lines. (please see the plots in Supplementary Fig.S1). As expected, BAY1816032 did not inhibit cell growth in these cells wherein BUB1 was silenced. In contrast, BAY1816032 significantly decreased cell viability in H1975 cells transfected with control siRNA. However, the reduction in control siRNA-transfected A549 cells was not significant in this experiment.

4) A caveat of the paper is that it does not provide any insight into the signaling pathways underlying the growth-suppressive activity of BAY1816032, cisplatin and radiation. They have shown that BUB1 inhibition DNS strand break repair and promotes programmed cell death. They need to show the cytoplasmic signaling network at which these there converge and how that phenomenon causes enhanced growth-suppressive activity in lung cancer cells.

This study was aimed to assess the potential synergistic effects of BAY1816032 (a BUB1 inhibitor) with established therapies including a PARP inhibitor (olaparib), platinum agent (cisplatin), and microtubule depolymerization inhibitor (paclitaxel). The results presented here demonstrate that tested combinations are synergistic. In future, we will validate these results in in-vivo preclinical studies and will focus on identifying the mechanisms. We anticipate that the combination of BUB1i with radiotherapy and olaparib, cisplatin and paclitaxel will impair cell-cycle, cause mitotic catastrophe, potentially affect DNA double strand break repair through NHEJ or HR pathways and increase apoptotic cell death (cartoon of potential mechanism is presented in new Figure 9). Please see our response to Reviewer #1, comment #4.

5) The PI observed that "pretreatment with BAY1816032 24 hours prior to cisplatin resulted in antagonism, which potentiated at higher cisplatin concentration. Similarly, pretreatment with cisplatin followed by BAY1816032 exhibited antagonism. Concurrent BAY1816032 and cisplatin led to synergism". The authors should comment on these observations in the DISCUSSION and provide a possible hypothesis why the combination of BAY1816032 and cisplatin sometimes led to antagonistic interactions and sometimes led to synergy.

Thank you for highlighting these important observations. We have now added a section within “Discussion” wherein we reasoned on these contradictory observations (see below and page 13, lines 392-413).

We hypothesize that antagonism during sequential treatment occurs because cisplatin induces DNA damage that arrests the cell cycle in the S and G2 phases, preventing cells from entering mitosis, where BUB1 is essential. Since BUB1 inhibition targets the spindle assembly checkpoint during mitosis, its effectiveness is diminished when the cell cycle is halted earlier by cisplatin. Similarly, if BUB1 inhibition is applied first, it may stall cells in a phase where cisplatin is less effective leading to antagonism. Additionally, BAY1816032 may influence DNA damage response pathways, particularly non-homologous end joining (NHEJ), thereby reducing the efficacy of subsequent cisplatin treatment, and contributing to antagonism. In contrast, the synergy observed with concurrent treatment likely results from the simultaneous induction of DNA damage by cisplatin and mitotic errors by BUB1 inhibition, overwhelming the cells and leading to enhanced cell death. Concurrent administration may allow both agents to act on different but complementary cellular targets simultaneously, resulting in increased synergy. Thus, the timing and sequence of drug administration are crucial in determining the interaction between BAY1816032 and cisplatin. This hypothesis will be further explored in the future to optimize combination therapies involving BUB1 inhibitors.

Reviewer 3 Report

Comments and Suggestions for Authors

The study focused on the effects of BUB1 inhibitor in sensitizing lung cancer cells to standard of care treatments in the clinic. The authors first evaluated the relevance of BUB1 in datasets such as the TCGA, and found potential for p53 in BUB1 regulation and synthetic lethal partner. Next, the authors tested the tolerated dose for normal vs lung cancer cell lines, and demonstrated that BUB1 inhibitor dose in normal cells is 4-10 times higher than those in cancer cells. Importantly, this is the first study showing that BUB1 inhibition can sensitize cancer cells to chemo and radio therapies. Overall, the low toxicity in normal cells and the synergistic effects of BUB1 inhibitor supports the high potential of BUB1 inhibitor in combination treatment regimens with standard of care. 

The article could improve by addressing the following: 

1. The study utilized a panel of lung cancer cell lines with both Mutant and wild type p53 status, did the authors consider overexpressing or knocking in p53 to wildtype or normal lung cell lines to further explore the mechanistic connection between p53 and BUB1?

2. What reason(s) do the authors consider to be the cause of differential results between concurrent use of BUB1 inhibitor and cisplatin and pretreatment/ at low BUB1 inhibitor doses? If BUB1 inhibitor functions to prolong gamma H2AX, and cisplatin also functions through DNA damage, why is there antagonistic effects with pretreatment? 

Author Response

Reviewer 3:

Comments and Suggestions for Authors

The study focused on the effects of BUB1 inhibitor in sensitizing lung cancer cells to standard-of-care treatments in the clinic. The authors first evaluated the relevance of BUB1 in datasets such as the TCGA and found potential for p53 in BUB1 regulation and synthetic lethal partner. Next, the authors tested the tolerated dose for normal vs lung cancer cell lines and demonstrated that the BUB1 inhibitor dose in normal cells is 4-10 times higher than those in cancer cells. Importantly, this is the first study showing that BUB1 inhibition can sensitize cancer cells to chemo and radio therapies. Overall, the low toxicity in normal cells and the synergistic effects of BUB1 inhibitor supports the high potential of BUB1 inhibitor in combination treatment regimens with standard of care. 

The article could improve by addressing the following: 

We thank reviewer #3 for their thorough evaluation of our manuscript and for the valuable suggestions that have significantly improved the quality of our work.

  1. The study utilized a panel of lung cancer cell lines with both Mutant and wild type p53 status, did the authors consider overexpressing or knocking in p53 to wildtype or normal lung cell lines to further explore the mechanistic connection between p53 and BUB1?

Thank you for the insightful suggestion. Although we have not overexpressed wt/mutant p53 in the present studies, we do plan to focus on this signaling axis in the future. Our future work will aim to investigate these interactions in greater detail, including manipulating p53 expression in relevant cell models to better understand its regulation of BUB1 and its broader implications for lung cancer treatment. Please see the response to Reviewer #1, comment #4.

  1. What reason(s) do the authors consider to be the cause of differential results between concurrent use of BUB1 inhibitor and cisplatin and pretreatment/ at low BUB1 inhibitor doses? If BUB1 inhibitor functions to prolong gamma H2AX, and cisplatin also functions through DNA damage, why are there antagonistic effects with pretreatment?

Please see our response to reviewer #2, point #5 above. We think that cell line responses are based on the phase of the cell-cycle when they are treated with BUB1 inhibitor or cisplatin and the phase cells are arrested in when they are exposed to the second inhibitor/drug. We do plan to find the underlaying mechanism in future studies.